# Metabolite Production in *Alkanna tinctoria* Links Plant Development with the Recruitment of Individual Members of Microbiome Thriving at the Root-Soil Interface

Cintia Csorba,[a] Nebojša Rodić,[b] Yanyan Zhao,[c] Livio Antonielli,[a] Günter Brader,[a] Angeliki Vlachou,[b] Evangelia Tsiokanos,[d] Ismahen Lalaymia,[c] Stéphane Declerck,[c] Vassilios P. Papageorgiou,[b] Andreana N. Assimopoulou,[b] Angela Sessitsch[a]

aCenter for Health & Bioresources, Bioresources Unit, AIT Austrian Institute of Technology GmbH, Tulln, Austria

bSchool of Chemical Engineering, Laboratory of Organic Chemistry and Center of Interdisciplinary Research and Innovation of AUTh, Natural Products Research Centre of Excellence (NatPro-AUTh), Aristotle University of Thessaloniki, Thessaloniki, Greece

cEarth and Life Institute, Mycology, Université Catholique de Louvain, Louvain-la-Neuve, Belgium

dFaculty of Pharmacy, Department of Pharmacognosy and Natural Product Chemistry, National and Kapodistrian University of Athens, Athens, Greece

Cintia Csorba, Nebojša Rodić, and Yanyan Zhao contributed equally to this work. Cintia Csorba is listed first as she was responsible for the coordination of analysis and writing; other cofirst authors follow in alphabetical order.

**ABSTRACT** Plants are naturally associated with diverse microbial communities, which play significant roles in plant performance, such as growth promotion or fending off pathogens. The roots of *Alkanna tinctoria* L. are rich in naphthoquinones, particularly the medicinally used enantiomers alkannin and shikonin and their derivatives. Former studies already have shown that microorganisms may modulate plant metabolism. To further investigate the potential interaction between *A. tinctoria* and associated microorganisms, we performed a greenhouse experiment in which *A. tinctoria* plants were grown in the presence of three distinct soil microbiomes. At four defined plant developmental stages, we made an in-depth assessment of bacterial and fungal root-associated microbiomes as well as all extracted primary and secondary metabolite content of root material. Our results showed that the plant developmental stage was the most important driver influencing the plant metabolite content, revealing peak contents of alkannin/shikonin derivatives at the fruiting stage. Plant root microbial diversity was influenced both by bulk soil origin and to a small extent by the developmental stage. The performed correlation analyses and cooccurrence networks on the measured metabolite content and the abundance of individual bacterial and fungal taxa suggested a dynamic and at times positive or negative relationship between root-associated microorganisms and root metabolism. In particular, the bacterial genera *Labrys* and *Allorhizobium-Neorhizobium-Pararhizobium-Rhizobium* as well as four species of the fungal genus *Penicillium* were found to be positively correlated with higher content of alkannins.

**IMPORTANCE** Previous studies have shown that individual, isolated microorganisms may influence secondary metabolism of plants and induce or stimulate the production of medicinally relevant secondary metabolism. Here, we analyzed the microbiome-metabolome linkage of the medicinal plant *Alkanna tinctoria*, which is known to produce valuable compounds, particularly the naphthoquinones alkannin and shikonin and their derivatives. A detailed bacterial and fungal microbiome and metabolome analysis of *A. tinctoria* roots revealed that the plant developmental stage influenced root metabolite production, whereas soil inoculants from three different geographical origins in which plants were grown shaped root-associated microbiota. Metabolomes of plant roots of the same developmental stage across different soils were highly similar, pinpointing to plant maturity as the primary driver of secondary metabolite production. Correlation and network analyses identified bacterial and fungal taxa showing a positive relationship between root-associated microorganisms and root metabolism. In particular,

Address correspondence to Angela Sessitsch, angela.sessitsch@ait.ac.at.

The authors declare no conflict of interest.

the bacterial genera *Allorhizobium-Neorhizobium-Pararhizobium-Rhizobium* and *Labrys* as well as the fungal species of genus *Penicillium* were found to be positively correlated with higher content of alkannins.

**KEYWORDS** plant, microbiome, alkannin, alkanet, metabolomics, metabolome, medicinal plant

Medicinal plants represent a valuable source of new drugs due to the production of bio-active secondary metabolites (SMs) (1–4). Among them, members of the *Boraginaceae* family, which comprise more than 150 genera and approximately 2,700 species worldwide, represent a valuable source of bioactive SMs, primarily naphthoquinones, alkaloids, flavonoids, polyphenols, phytosterols, and terpenoids (5). In particular, the naphthoquinone alkannin, its enantiomer shikonin, and their derivatives (A/S) are commercially interesting due to their broad spectrum of biological activities, such as wound healing (6, 7), anti-inflammatory (8, 9), and anticancer (10) activities.

Environmental factors, developmental stage, light intensity, soil composition, or various abiotic stress factors have been described to qualitatively and quantitatively influence primary and secondary metabolism of plants (11, 12). For example, the root and rhizome of 2-year-old *Echinacea purpurea* M. was reported to produce larger amounts of the phenylpropanoid cichoric acid in the fruiting stage (13), while the main flavonoids in the roots of *Scutellaria baicalensis* G. accumulated rapidly before the full-bloom stage (14).

Plants are associated with highly diverse microbial communities composed primarily of bacteria and fungi and termed as the plant microbiome. The microbiome plays a key role for plant health, development, and productivity by mobilizing plant nutrients, increasing nutrient uptake, or antagonizing plant pathogens and by enhancing plant resilience to abiotic stresses (15–17). Some microorganisms release phytohormones, small molecules, or volatile compounds that may act directly or indirectly to activate plant immunity or regulate plant growth and morphogenesis (18). A number of studies also reported plant-associated microorganisms modulating plant metabolite synthesis (19, 20) and influencing the quantity and quality of the produced SMs (21). For example, inoculation of *Chamomilla recutita* L. with a *Bacillus subtilis* and a *Paenibacillus polymyxa* strain (21) and coinoculation of *Papaver somniferum* L. with *Acinetobacter* sp. and *Marmoricola* sp. (22) resulted in the enhancement of specific bioactive SMs (apigenin-7-O-glucoside and morphine, respectively). Furthermore, endophytic bacteria isolated from *Echinacea purpurea* L. have been reported to stimulate changes in the expression of plant valine decarboxylase (VDC) genes for SM alkamide biosynthesis (13). Also, fungi have been reported to enhance SM production in plants. Yams inoculated with different arbuscular mycorrhizal fungal (AMF) species (23) and *Rumex gmelini* T. seedlings cocultured with *Aspergillus* sp. isolated from the roots of tissue culture seedlings (24) showed significantly enhanced production of bioactive SMs in *R. gmelini*.

*Alkanna tinctoria* (L.) Tausch, commonly known as alkanet, is a European perennial flowering plant of the *Boraginaceae* family and an important producer of A/S. The derivatives of alkannin include mainly acetylalkannin, propionylalkannin, isobutylalkannin, angelylalkannin, isovalerylalkannin, $\alpha$-methyl-$n$-butylalkannin, $\beta$-hydroxy-isovalerylalkannin, and $\beta,\beta$-dimethylacrylalkannin as well as others (25). *A. tinctoria* was an ideal naphthoquinone-producing candidate for our study as it has a high A/S content (9, 26), it is endemic in Europe, and good quality clonal material is available. Recently, it was shown that A/S production can be enhanced by the application of microorganisms. *A. tinctoria* hairy roots inoculated with four bacterial strains belonging to *Chitinophaga* sp., *Allorhizobium* sp., *Duganella* sp., and *Micromonospora* sp., all isolated from wild *A. tinctoria* roots, showed a significantly higher production of A/S than the uninoculated controls (27). However, the mechanisms leading to this enhanced SM production are unknown. Similarly, the ecology of plant-associated microbiomes of *A. tinctoria* or their link with plant SM production have so far not been investigated. Furthermore, although protocols for untargeted

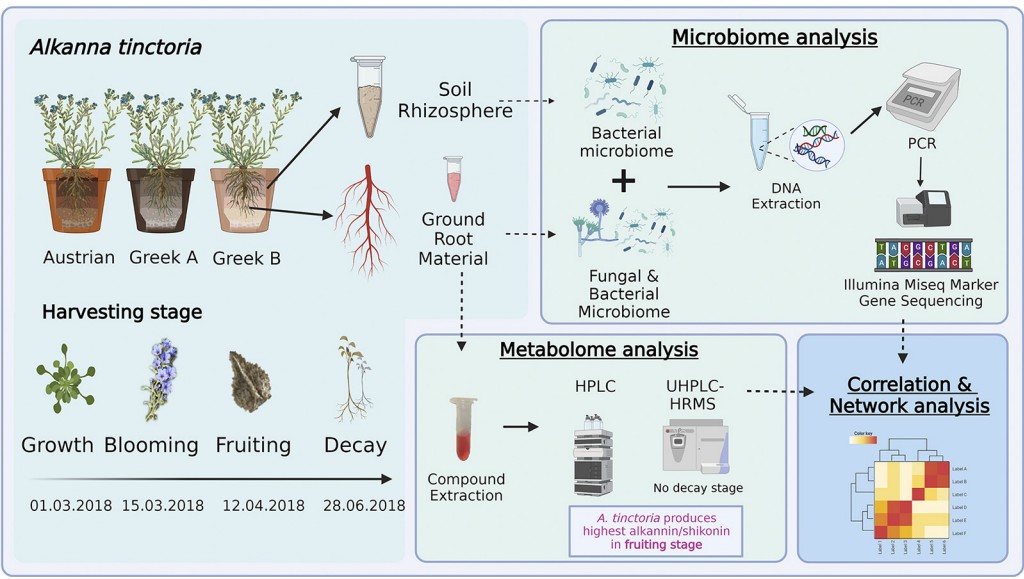

**FIG 1** Graphical overview of the experiment. Created with BioRender.com.

plant metabolomics analysis by using ultrahigh performance liquid chromatography high-resolution mass spectrometry (UHPLC-HRMS) have been published (28), holistic studies of *Boraginaceae* metabolomes are still missing. Here, we present a comprehensive metabolomics study on a *Boraginaceae* species in which we utilize UHPLC-HRMS for analyzing the metabolome of *A. tinctoria* roots.

Although secondary metabolite production is mostly driven by the plant and its physiology, it is our hypothesis that the plant microbiome (besides other environmental parameters) may modulate secondary metabolism. Root-associated microorganisms may influence the production of A/S and other SMs of *A. tinctoria* roots, and the plant metabolome is also likely to affect the microbiome structure. Consequently, we expected that the root microbiome or specific taxa would correlate with the overall metabolome or specific plant SMs. Furthermore, our hypothesis was that different A/S concentrations are produced in different plant growth stages, in line with a growth stage-dependent colonization of plant microbiota. To test our hypotheses, we performed next-generation sequencing of the root-associated bacterial and fungal microbiome of *A. tinctoria* grown for 5 months in the greenhouse in the presence of three different soil microbiomes and correlated the presence of individual bacterial and fungal taxa with the metabolic content results of HPLC and UHPLC-HRMS performed on the root extracts (Fig. 1). Apart from addressing A/S and naphthoquinone compounds, we also performed a comprehensive, untargeted metabolome analysis of *A. tinctoria*, which was used in microbiome-metabolome correlations and cooccurrence network analyses.

## RESULTS

**Root bacterial and fungal microbial diversity and taxonomic composition.** We focused on the analysis of the root-associated bacterial and fungal microbiome; in addition we report on the bacterial microbiome of rhizosphere and bulk soil samples. Analysis of variance (ANOVA) analyses on alpha-diversity estimators (Simpson's diversity and richness) in root samples (comprising endophytic and rhizoplane microorganisms) showed that the soil microbiome was the only significant factor among the analyzed components (Table 1 and Fig. S1 in the supplemental material). As we inoculated 200 g of nonsterile soil in 5 kg of sterilized soil substrate, we consider any observed differences between soils mostly due to the different microbiomes present in these soils. In the rhizosphere, both soil (microbiome) and developmental stage significantly affected the alpha-diversity values (bacterial diversity and richness; Table 1), whereas in bulk soil samples, only the plant developmental stage influenced alpha-diversity. According to pairwise permutational multivariate analysis

**TABLE 1** Alpha-diversity statistics

| Sample type | Bacteria Factor | Richness[a] | | Diversity[a] | |
|---|---|---|---|---|---|
| | | *F* value | *P* value[b] | *F* value | *P* value[b] |
| **Root** | Soil[c] | 2.76 | 0.071 | 5.46 | **0.007**\*\* |
| | Stage | 2.74 | 0.051 | 0.51 | 0.678 |
| | Soil:Stage | 2.04 | 0.075 | 0.52 | 0.794 |
| **Rhizosphere** | Soil[c] | 5.92 | **0.004**\*\* | 47.09 | **0.000**\*\*\* |
| | Stage | 23.48 | **0.000**\*\*\* | 17.31 | **0.000**\*\*\* |
| | Soil:Stage | 2.95 | **0.014**\* | 15.52 | **0.000**\*\*\* |
| **Soil** | Soil[c] | 1.55 | 0.221 | 2.40 | 0.099 |
| | Stage | 6.99 | **0.000**\*\*\* | 12.33 | **0.000**\*\*\* |
| | Soil:Stage | 2.71 | **0.021**\* | 1.26 | 0.289 |
| Sample type | Fungi Factor | Richness | | Diversity | |
| | | *F* value | *P* value | *F* value | *P* value |
| **Root** | Soil[c] | 5.39 | **0.007**\*\* | 4.47 | **0.015**\* |
| | **Stage** | 1.77 | 0.161 | 2.07 | 0.113 |
| | **Soil:Stage** | 1.25 | 0.294 | 0.53 | 0.781 |

| Bacteria[d] | | | Fungi | | |
|---|---|---|---|---|---|
| Growth stage | Richness values[d] | Diversity values[d] | Growth stage | Richness values | Diversity values |
| **Austrian** | 157 | 0.974 | **Austrian** | 31.9 | 0.677 |
| **Greek A** | 165 | 0.979 | **Greek A** | 39.8 | 0.789 |
| **Greek B** | 141 | 0.959 | **Greek B** | 43.7 | 0.782 |
| **Blooming** | | | **Blooming** | | |
| **Austrian** | 134 | 0.966 | **Austrian** | 28.6 | 0.642 |
| **Greek A** | 142 | 0.971 | **Greek A** | 36.6 | 0.763 |
| **Greek B** | 118 | 0.952 | **Greek B** | 40.4 | 0.747 |
| **Fruiting** | | | **Fruiting** | | |
| **Austrian** | 127 | 0.967 | **Austrian** | 38.3 | 0.764 |
| **Greek A** | 135 | 0.973 | **Greek A** | 46.2 | 0.885 |
| **Greek B** | 111 | 0.953 | **Greek B** | 50.1 | 0.869 |
| **Decay** | | | **Decay** | | |
| **Austrian** | 150 | 0.966 | **Austrian** | 32.8 | 0.703 |
| **Greek A** | 158 | 0.971 | **Greek A** | 40.8 | 0.885 |
| **Greek B** | 134 | 0.952 | **Greek B** | 44.7 | 0.808 |

[a]ANOVA (Analysis of variance) based on linear model fitted to richness (observed number of ASVs) and Simpson's diversity values (SDI).
[b]Statistically significant *P* values presented in bold letters, \*, $P < 0.05$; \*\*, $P < 0.01$; \*\*\*, $P < 0.001$.
[c]Soil refers to soil communities of different origin.
[d]For root-associated bacteria and fungi also the estimated mean values without statistics per group are provided.

of variance (PERMANOVA) calculations on the Bray-Curtis dissimilarity values (Table S1), the bacterial beta-diversity of root samples in the different soil microbiome subsets was significantly different only between the fruiting and decay stages. There was only one significant difference in beta-diversity values between developmental stages in the root fungal microbiome, which was in the Greek B soil between blooming and fruiting stages (Table S1). However, the different soil microbiomes had significant effects on the root bacterial and fungal beta-diversity values after reaching the blooming stage (Table S1 and Fig. 2). In the rhizosphere and bulk soil samples, the measured beta-diversity values were significantly influenced by both developmental stage and soil microbiome (Fig. 2), although the difference between plant developmental stages varied in the different soil subsets (Table S1). Both rhizosphere and bulk soil bacterial communities exhibited significant differences in their composition between Austrian and Greek soils in all stages, except in the fruiting stage in which Austrian and Greek A groups were similar (Table S1). The most abundant bacterial phyla assigned from 2,261 amplicon sequence variants (ASVs) in all root samples were *Proteobacteria* (~67% relative abundance), *Bacteroidetes* (~16% relative abundance), and *Actinobacteria* (~11% relative abundance). The bacterial classes with the highest abundance in all root samples were *Gammaproteobacteria* (~44% relative abundance),

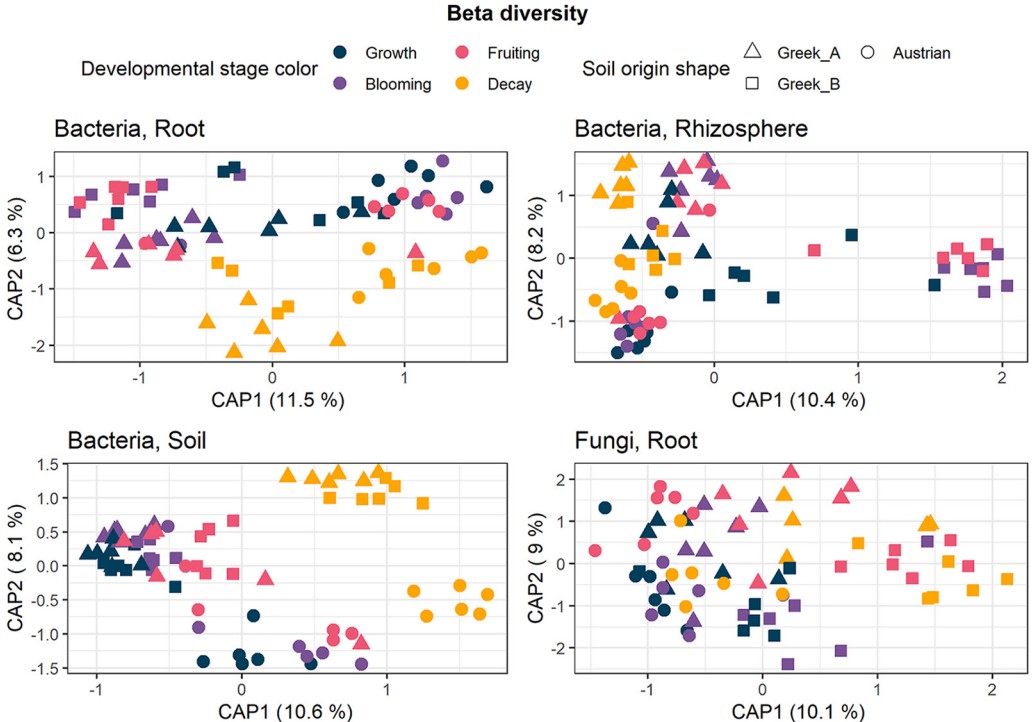

**FIG 2** CAP beta-diversity plots based on Bray-Curtis dissimilarities showing the differences in the microbial community structures (*n* = 6 constrained by the soil origin and by the different developmental stages in root, rhizosphere, and soil bacterial samples as well as in root fungal samples).

*Alphaproteobacteria* (~23% relative abundance), *Bacteroidia* (~16% relative abundance), and *Actinobacteria* (~10% relative abundance). The bacterial genus groups with the highest relative abundances in all stages and soils were *Burkholderia-Caballeronia-Paraburkholderia* and *Allorhizobium-Neorhizobium-Pararhizobium-Rhizobium* (Fig. 3A). Root fungal ASVs mostly belonged to the phylum *Ascomycota* (97% relative abundance) and to the class *Sordariomycetes* (84% relative abundance). The fungal genus *Thielaviopsis* dominated the vegetative growth (61% relative abundance) and blooming (73% relative abundance) stages of roots of plants grown in the presence of the Austrian soil amendment (Fig. 3B). The fungal genus composition of *A. tinctoria* roots grown in Greek A soil microbiome had a dynamic turnover throughout plant development, although the genus *Penicillium* was the most abundant in the fruiting stage in all soils (Fig. 3B).

In the rhizosphere, the genera *Burkholderia-Caballeronia-Paraburkholderia* were most abundant in the fruiting stage (34%), followed by *Mucilaginibacter* (only in the growth and decay stage; 6% relative abundance) and *Allorhizobium-Neorhizobium-Pararhizobium-Rhizobium* (5% relative abundance). In bulk soil samples, the genera *Burkholderia-Caballeronia-Paraburkholderia* were most abundant (22% relative abundance), followed by *Mucilaginibacter* (11% relative abundance), *Nocardioides* (9% relative abundance), *Dyella* (6% relative abundance), and *Gryllotalpicola* (5% relative abundance).

For each sample type, we identified core microbiomes composed of shared reproducibly occuring ASVs (rASVs) across all soils or developmental stages as well as stable core microbiomes of shared rASVs across soils and developmental stages. Among root-associated bacteria, 204 core rASVs were found in all root samples irrespective of the soil, whereas 193 rASVs were shared between all plant developmental stages. We found 174 stable core rASVs shared among all samples (Fig. S2A) comprising 9 phyla, 11 classes, and 57 genera and dominated by *Burkholderia-Caballeronia-Paraburkholderia* (36% relative abundance among core genera; Fig. S2B). In the root-associated fungal core microbiome, 59 rASVs were found in all root samples irrespective of the soil and 61 rASVs in all developmental stages, and we

**A**

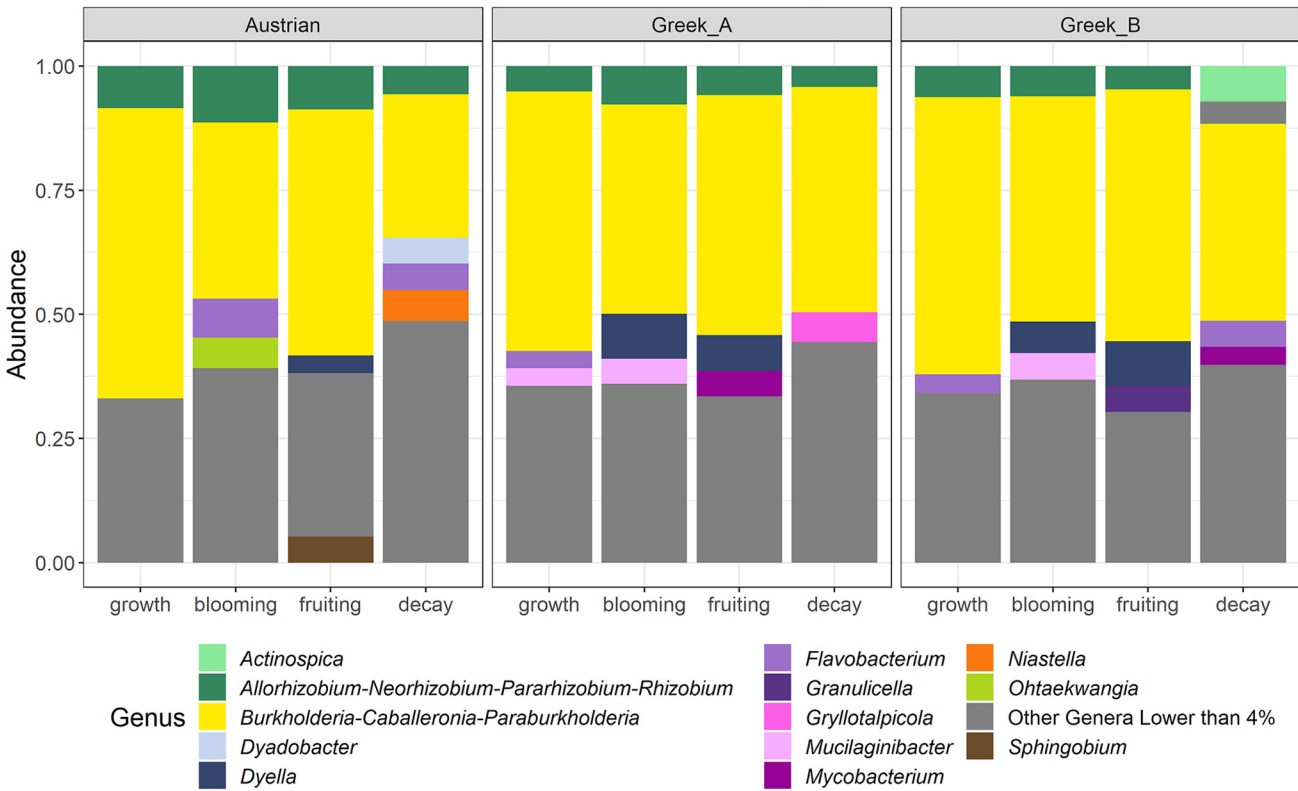

**B**

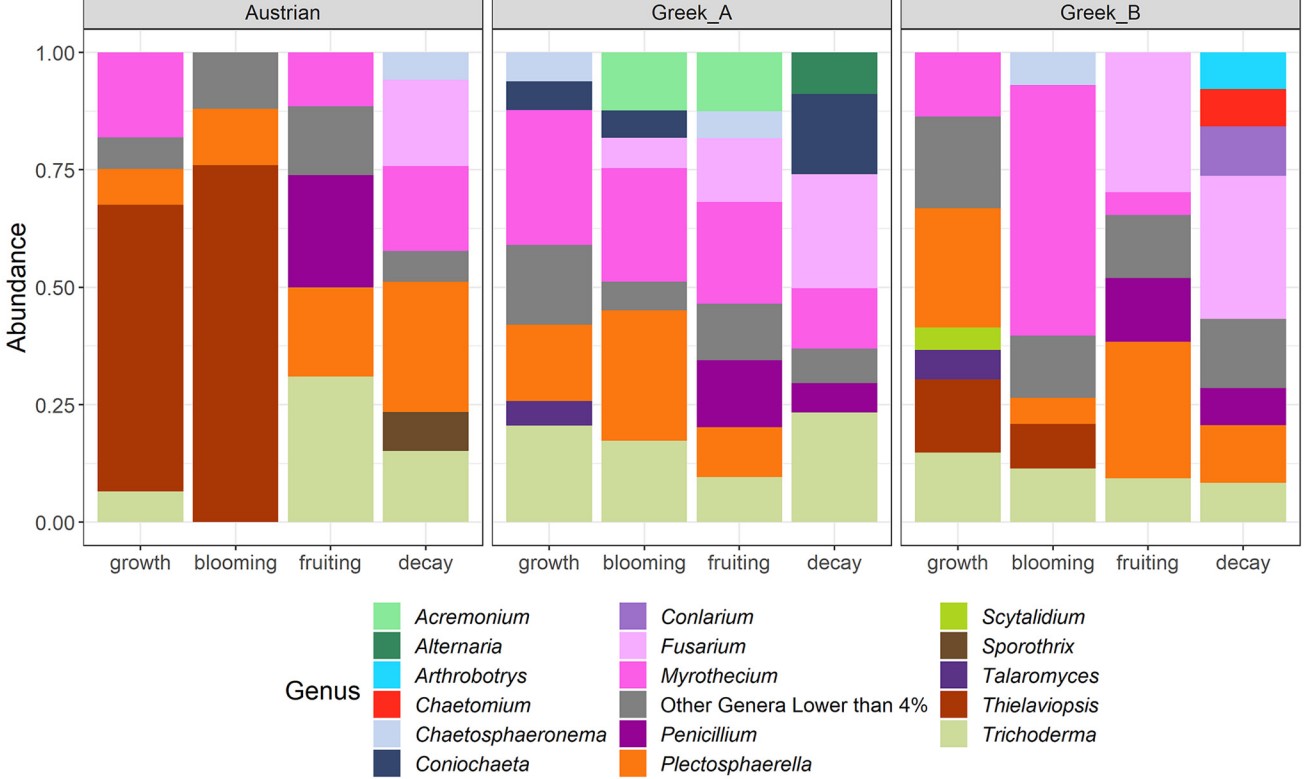

**FIG 3** Changes in relative abundance of genera in root samples during plant development. (A and B) Bacteria (A) and fungi (B) in the root samples of *A. tinctoria* faceted by the three soil microbial communities. Genera lower than 4% were merged.

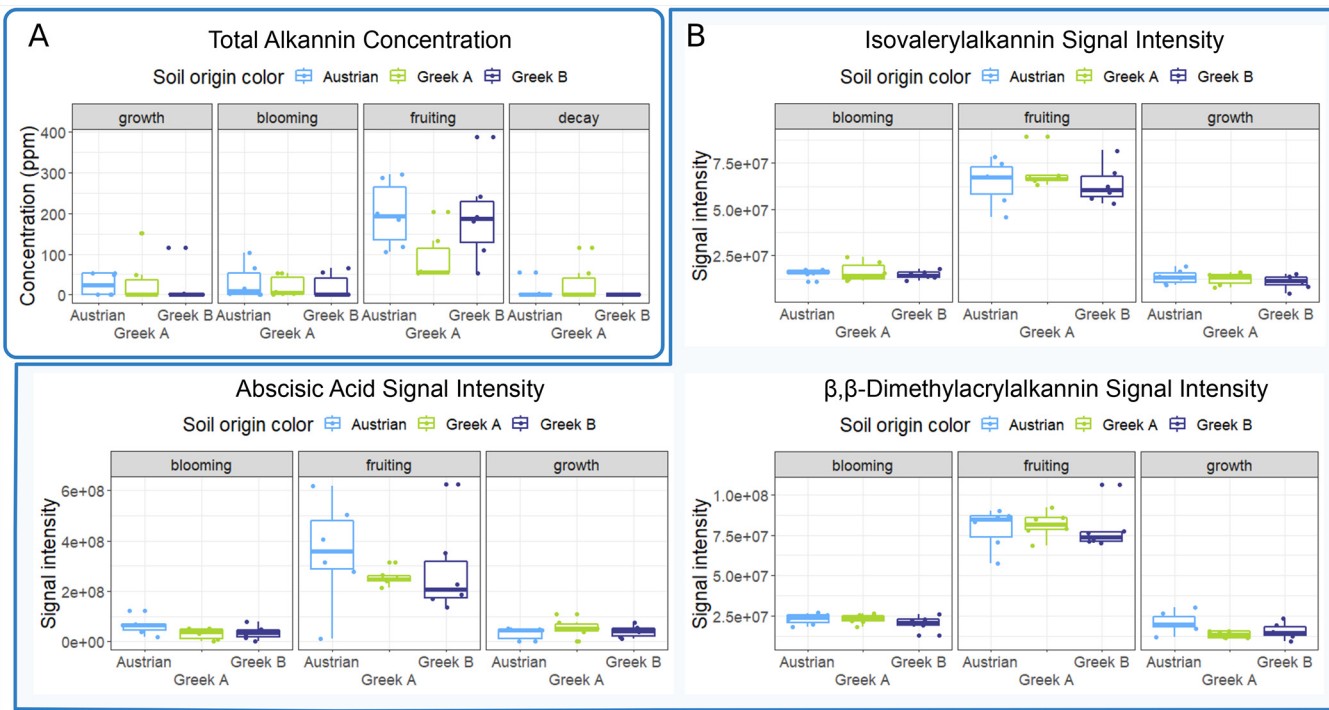

**FIG 4** (A) Total A/S concentration (mg/L) measured by HPLC in the methanolic extracts of *A. tinctoria* roots across four different growth stages and three different soil microbiomes. (B) UHPLC-HRMS signal intensities of important metabolites in the extracts of *A. tinctoria* roots across three different growth stages and three different soil microbial communities. Created with BioRender.com.

identified 56 stable core rASVs. The latter could be assigned to 2 phyla, 7 classes, and 22 genera and was dominated by *Trichoderma* (19% relative abundance), *Plectosphaerella* (19% relative abundance), *Fusarium* (16% relative abundance), and *Myrothecium* (15% relative abundance; Fig. S3).

The majority of the core bacterial rASVs in rhizosphere samples belonged to *Burkholderia-Caballeronia-Paraburkholderia* (69% relative abundance among all rASVs) and *Asticcacaulis* (10% relative abundance). The bulk soil core rASVs consisted mainly of *Nocardioides* (21% relative abundance), *Burkholderia-Caballeronia-Paraburkholderia* (19% relative abundance), and *Dyella* (13% relative abundance). Overall, 114 bacterial reproducibly occurring core rASVs were shared between root, rhizosphere, and bulk soil samples (Fig. S4) and belonged mainly to *Burkholderia-Caballeronia-Paraburkholderia*.

**A. tinctoria roots produce highest A/S amounts in the fruiting stage.** To determine the *A. tinctoria* developmental stage in which the roots exhibit the highest A/S content, a high-throughput gradient-elution HPLC method was applied. External calibration with commercially available and purified standards allowed the quantitation of individual A/S compounds as well as their sum, that is, "Total A/S."

The measured root A/S concentrations were highest in the fruiting stage, followed by the vegetative growth and blooming stages (Table S2). In the decay stage, roots contained the lowest A/S amounts, whereas in the fruiting stage, A/S total content of roots represented up to 1.7% of the total root weight based on methanolic extractions. The A/S content variability was high across all sample groups (Fig. 4A). Changes in total A/S content were primarily driven by the developmental stages (ANOVA, $P < 0.01$), while the variance due to different soil types was found to be statistically insignificant ($P > 0.05$; Table 2). This trend was displayed also with concentrations of individual A/S compounds, except for alkannin and acetylalkannin. The latter compound showed to be influenced by the soil ($P = 0.02$). In contrast, the alkannin data set consisted mainly of nondetectable and nonquantifiable values. This is a consequence of the total A/S content being dominated by two compounds: $\beta,\beta$-dimethylacrylalkannin and isovalerylalkannin. Other A/S were mostly present in small amounts, often eluding reliable

**TABLE 2** Statistics on A/S concentrations obtained from HPLC data

| ANOVA[a] | | Alkannins/shikonins | Concn | |
|---|---|---|---|---|
| Compound | Factor | | $F$ value | $P$ value[b] |
| Total A/S | Soil | | 1.24 | 0.296 |
| | Stage | | 28.26 | 0.000*** |
| | Soil:stage | | 2.12 | 0.063 |
| Alkannin | Soil | | 3.91 | 0.026* |
| | Stage | | 0.00 | 1 |
| | Soil:stage | | 0.00 | 1 |
| Acetylalkannin | Soil | | 3.94 | 0.024* |
| | Stage | | 16.07 | 0.000*** |
| | Soil:stage | | 2.29 | 0.047* |
| Deoxyalkannin | Soil | | 0.24 | 0.784 |
| | Stage | | 9.56 | 0.000*** |
| | Soil:stage | | 0.19 | 0.977 |
| Propionylalkannin | Soil | | 0.39 | 0.679 |
| | Stage | | 14.69 | 0.000*** |
| | Soil:stage | | 0.50 | 0.804 |
| $\beta,\beta$-Dimethylacrylalkannin + isovalerylalkannin | Soil | | 1.18 | 0.314 |
| | Stage | | 26.63 | 0.000*** |
| | Soil:stage | | 2.11 | 0.065 |
| **PERMANOVA[a]** | **Factor** | | $F$ **value** | $P$ **value**[b] |
| UHPLC-HRMS metabolome | Soil | | 1.33 | 0.133 |
| | Stage | | 11.97 | 0.000*** |

[a]ANOVA based on a linear model. PERMANOVA on the obtained UHPLC-HRMS data.
[b]*, $P < 0.05$; **, $P < 0.01$; ***, $P < 0.001$.

**HPLC detection/quantitation.** The list of annotated compounds identified in the greenhouse *A. tinctoria* root samples by using UHPLC-HRMS is shown in Table S3.

Chiral HPLC measurements were used for determining the enantiomeric ratio of A/S in selected root samples, which was shown to remain stable regardless of the developmental stage. The type of soil microbiome used in our study did not influence the enantiomeric ratio. Analyzed *A. tinctoria* roots contained, on average, ~95% alkannin and ~5% shikonin.

**Untargeted UHPLC-HRMS-based metabolomics revealed developmental stage as the dominant driver of metabolome changes.** A high-throughput UHPLC-HRMS method was used to analyze samples from the first three developmental stages. The untargeted UHPLC-HRMS metabolomics data set was used for metabolite identification and annotation.

We identified 4 A/S compounds and annotated 27 metabolites from the UHPLC-HRMS data set. Among these were compounds that belong to the alkannin biosynthesis pathway (29, 30), which was the main focus of our study. Plant hormones (abscisic acid, methyl jasmonate, and 3-methylsalicylic acid) were also detected in the data set. Other highlighted annotated metabolites included sinapinic acid, caffeic acid 2-amino-1,3,4-octadecanetriol or phytosphingosine, 6-methoxyflavanone, ostruthol, and compounds belonging to the anthraquinone class (a complete list of identified features is shown in Table S4). Moreover, we were able to detect citrinin, a mycotoxin from *Penicillium*.

In UHPLC-HRMS, isovalerylalkannin and $\beta,\beta$-dimethyacrylalkannin substances could be separated, and the two most abundant A/S in *A. tinctoria* roots showed the highest signal intensities across different sample groups, with obvious maxima in the fruiting stage. Abscisic acid showed similar trends in signal intensities as the two A/S, reaching maximum concentrations in the same developmental stage, albeit with higher variability between individual samples (Fig. 4B). In a correlation analysis, abscisic acid was found to cluster with 6-methoxyflavanone, ostruthol, maltotetraose, and A/S (all metabolites present in high relative concentration in the fruiting stage). Another distinct cluster contained caffeic acid, rosmarinic acid, sinapinic acid, methyl jasmonate, and 4-(3,4-dihydroxyphenyl)-6,7-dihydroxy-2-naphthoic acid, metabolites that are present in high relative concentrations at the growth and blooming stages (Fig. 5A).

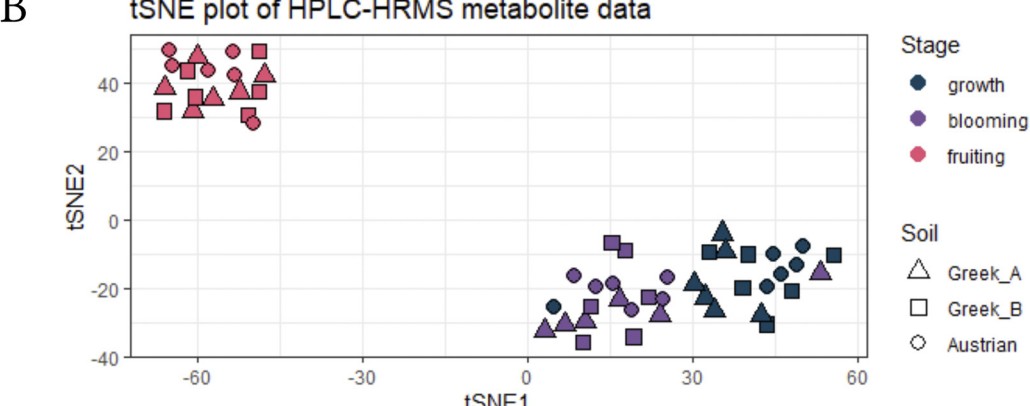

**FIG 5** (A) Compound correlation heatmap from the UHPLC-HRMS data set. DHPNA, 4-(3,4-dihydroxyphenyl)-6,7-dihydroxy-2-naphthoic acid; HMBA, 4-hydroxy-3-(3-methylbut-2-en-1-yl)benzoic acid; AODT, 2-amino-1,3,4-octadecanetriol; MPTA, methyl (7-hydroxy-11-methyl-2,9-dioxo-12,15-dioxatetracyclo[8.4.1.0¹,¹⁰.0³,⁸]pentadeca-3,5,7-trien-13-yl)acetate. (B) t-SNE dimensionality reduction plot on centered and scaled total UHPLC-HRMS data. Created with BioRender.com.

We combined ANOVA, PERMANOVA, and t-distributed stochastic neighbor embedding (t-SNE; Fig. 5B) to analyze the large untargeted metabolomics data sets obtained via UHPLC-HRMS. These indicated no significant variations in the metabolome of *A. tinctoria* roots in the three different soil substrates. PERMANOVA confirmed the t-SNE

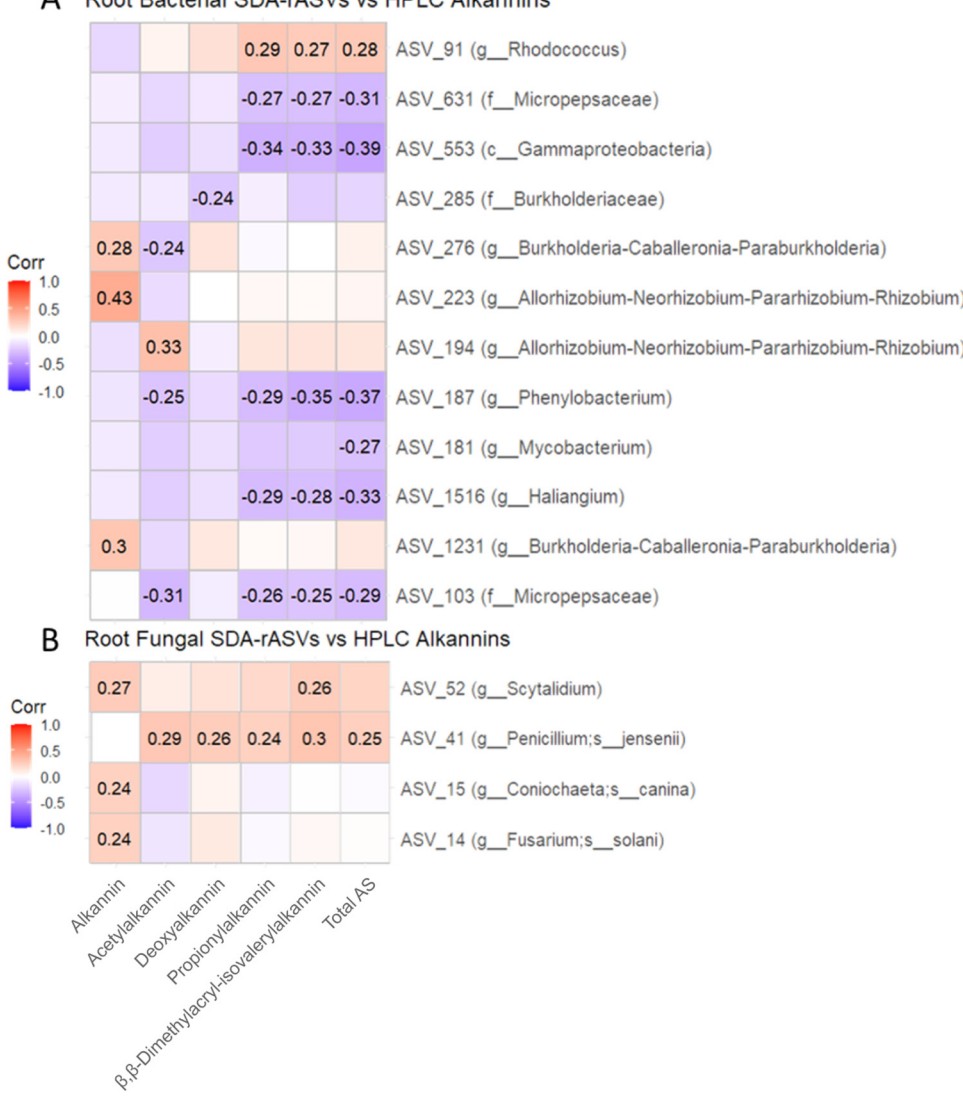

**FIG 6** (A and B) Correlation plots of Spearman correlation values calculated from the abundance of alkannin concentrations from HPLC analysis and significantly differentially abundant root bacterial rASVs (A) and significantly differentially abundant root fungal taxa (B). *R* values are written in the boxes on the plots, and positive or negative attribution of the correlations are indicated by the color legend. Nonsignificant correlations (*P* > 0.05) are without *R* values on the plot.

patterns on our data, and only the developmental stage was found to be a significant factor. When ANOVA was used to compare samples from different soil substrates (differing in their microbiomes) within the same developmental stage, only a single feature (that unfortunately eluded annotation) was statistically significant. ANOVA highlighted hundreds of statistically significant features in the UHPLC-HRMS data set when samples from the same soil microbiome type were compared across different developmental stages. Due to the lack of confident tandem mass spectrometry (MS/MS) spectral matches, the majority of these features did not result in compound annotation or identification. The t-SNE dimensionality reduction (Fig. 5B) also discriminated between the three different developmental stages.

**Individual bacterial and fungal taxa associated with roots positively correlated with higher levels of alkannins.** For correlating alkannin concentrations with root-associated microorganisms in our data set, we calculated significantly differentially abundant rASVs (sign. rASV) from *A. tinctoria* root samples. The identified 31 bacterial and 10 fungal sign. rASVs were present in at least 4 of 6 replicates in their individual group but showed

significantly different abundances between the groups (one group for each soil/developmental stage combination). We used these root sign. rASVs for the correlation analysis with the targeted HPLC root metabolite analysis results to identify possible links between A/S production and the presence of individual bacterial or fungal taxa. In a Spearman correlation between the root bacterial sign. rASVs and the concentrations of the targeted HPLC metabolites, rASV 223 had the strongest positive correlation ($R = 0.43$) with alkannin (Fig. 6A). This rASV was assigned to the genus group *Allorhizobium-Neorhizobium-Pararhizobium-Rhizobium*, which was present in all stages in roots but only when plants were cultivated in one of the Greek soils (Table S3). Root bacterial sign. rASV 194, from the same genus group, was positively correlated with acetylalkannin ($R = 0.33$) and was only present in samples grown in substrate amended with the Austrian soil. Bacterial sign. rASV 194 was also found both in soil and rhizosphere samples in the fruiting stage of all three soils and in all stages when plants were grown in the Austrian soil (Table S3). Additionally, bacterial sign. rASV 1231 of the genus group *Burkholderia-Caballeronia-Paraburkholderia* positively correlated with alkannin ($R = 0.3$).

The root fungal sign. rASV 41, assigned to *Penicillium jensenii*, positively correlated with $\beta,\beta$-dimethylacryl-isovalerylalkannin ($R = 0.30$; Fig. 6B). This sign. rASV was found in the fruiting stage in all root samples, with the highest abundance in the Greek B samples at the fruiting stage (Table S4).

We built cooccurrence networks from our root microbiome and UHPLC-HRMS data for association discovery. We created two networks from all multiomics data via performing two random forest analyses based on development stages or soil types on all data separately and then calculated Spearman correlation matrices based on the significant ($P < 0.05$) features. We used a higher threshold of the correlation coefficients ($R > 0.5$) in the creation of the adjacency matrices to have transparent networks.

The network based on the soil types (Fig. 7A) resulted in 700 total significant edges and 122 nodes. Key nodes in this network were bacterial ASV 54 (*Dyella* sp.) and bacterial ASV 194 (*Allorhizobium-Neorhizobium-Pararhizobium-Rhizobium*). In this network, the largest cliques consisted only of bacterial and fungal ASVs, which highlights the importance of microbial nodes and their connection in this scenario.

The developmental stage network (Fig. 7B) contained 25,541 total significant edges and 362 nodes. The key nodes in this network were metabolites "M352T3" and "M284T3_1," while the largest cliques consisted exclusively of metabolite nodes. M352T3 is likely senecivernine N-oxide, while the latter metabolite unfortunately eluded annotation.

Additionally, we created a filtered subnetwork based on the developmental stage random forest analysis with only the identified/annotated UHPLC-HRMS compounds (Fig. S5). We correlated those against the random forest-filtered root microbiome and used a threshold of $R > 0.3$ for building a more detailed, but still positively correlating, network. This subnetwork contained only 6 identified or annotated metabolites and in total contained 306 significant edges and 50 nodes. The key nodes/vertices of this network were isovalerylalkannin, which was also the node with the maximum betweenness, and bacterial ASV 294, an alphaproteobacterium from the genus *Labrys* with the highest eigenvector centrality in the network. Both of these important nodes were associated with the fruiting stage. The direct neighbors of isovalerylalkannin included positive correlations with alkannin, $\beta,\beta$-dimethylacryalkannin, abscisic acid, maltotetraose, and 6-methoxyflavanone. Direct fungal connections to isovalerylalkannin were ASV 39 (*Penicillium simplicissimum*, $R = 0.38$), ASV 41 (*Penicillium jensenii*, $R = 0.46$), ASV 56 (*Penicillium* sp., $R = 0.38$), ASV 85 (*Penicillium citreonigrum*, $R = 0.46$), and ASV 111 (*Saitozyma podzolica*, $R = -0.33$). The direct bacterial neighbors to isovalerylalkannin were ASV 18 (*Nocardioides* sp., $R = -0.35$), ASV 77 (*Conexibacter woesei*, $R = 0.38$), ASV 80 (*Micropepsaceae* sp., $R = -0.34$), ASV 156 (*Allorhizobium-Neorhizobium-Pararhizobium-Rhizobium*, $R = -0.36$), ASV 259 (*Asticcaulis* sp., $R = -0.41$), ASV 271 (*Sphingomonas* sp., $R = 0.34$), ASV 285 (*Burkholderiaceae* sp., $R = -0.31$), ASV 290 (*Chryseolinea* sp., $R = -0.35$), ASV 294 (*Labrys* sp., $R = 0.54$), ASV 407 (*Novosphingobium resinovorum*, $R = -0.40$), ASV 416 (*Rhodopseudomonas* sp., $R = 0.30$), ASV 424 (*Cellvibrio* sp., $R = -0.35$), and ASV 673 (*Methylophilaceae* sp., $R = -0.31$).

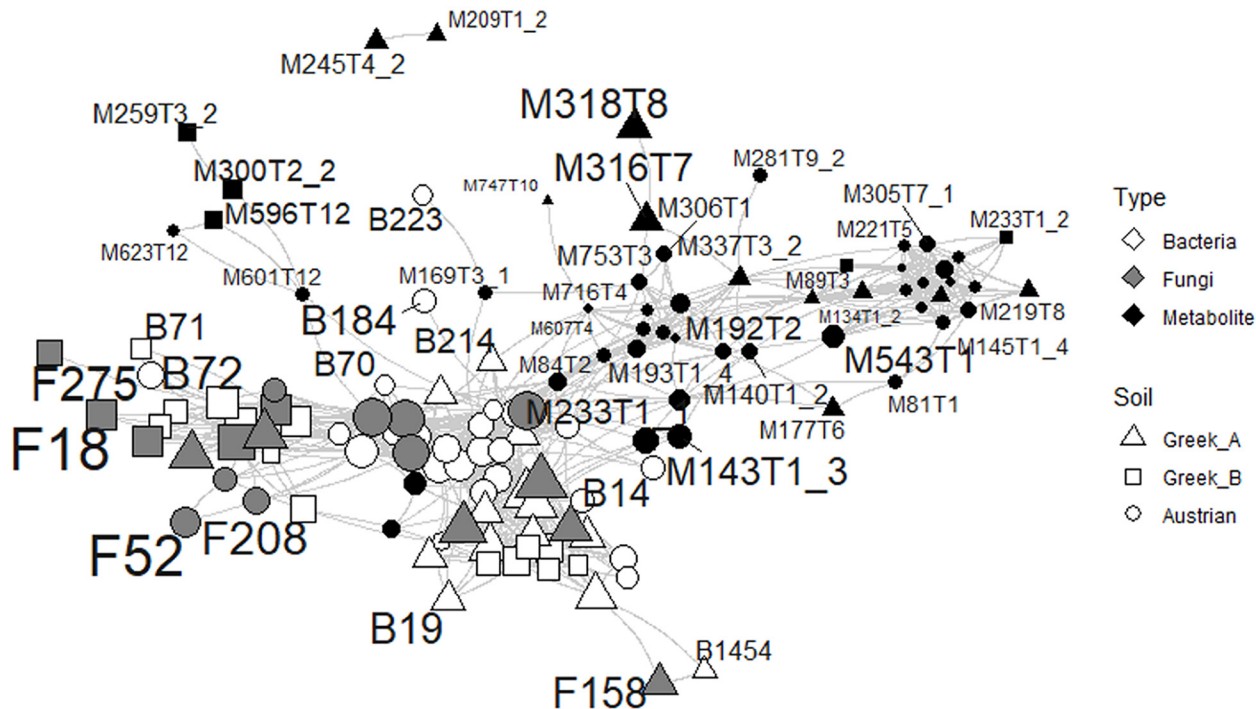

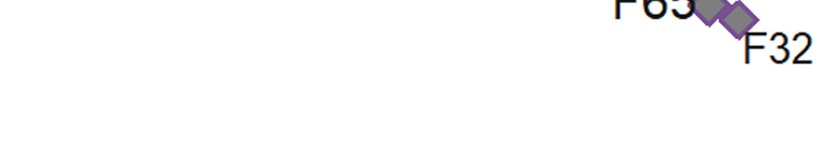

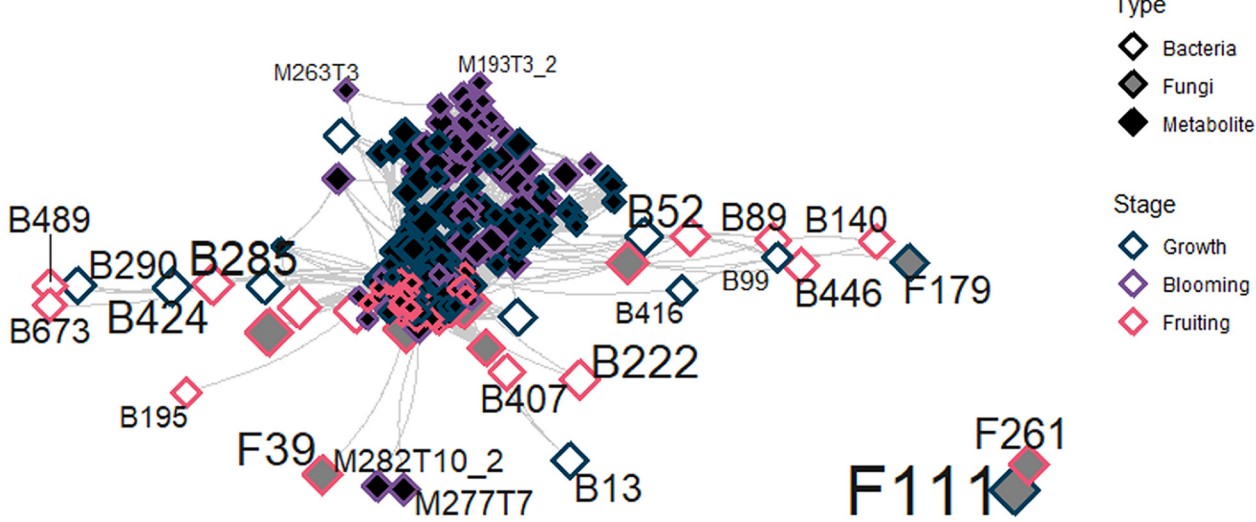

**FIG 7** Cooccurrence network of metabolites and microbial ASVs. (A) Cooccurrence network based on soil type differentiation; the key nodes of the network were B54 and B194. (B) Cooccurrence network based on developmental stages; the key nodes of the network were M352T3 and M284T3_1. Nodes starting with M are metabolites, B nodes are the bacterial rASVs, and F nodes are the fungal rASVs. Key nodes and labels are not visible on either network due to high node density in the central hubs.

## DISCUSSION

Several studies have focused on the microbiome and metabolome of medicinal plants with prominent SM content, although multidisciplinary approaches combining molecular microbiological and advanced chemical analytical methods are still rare. The focus of former studies in the topic has been restricted mainly to testing microbial isolates regarding their effect on plant secondary metabolism (22, 27, 31). Here, we combined next-generation sequencing of root-associated microbiomes of *A. tinctoria* with a multilayered metabolome analysis.

We identified the plant developmental stage as the most important driver of the root metabolome change, whereas the microbiome was, to a major extent, influenced by the soil with its specific microbiome. The influence of other soil parameters can be mostly ruled out, as plants were grown in a sterilized soil substrate, which was amended with a comparably small amount of nonsterile soils obtained from different geographic regions. In this study, the soils had very similar pH values; however, soil pH may strongly influence plant microbiomes and metabolism. In this study, metabolomes of plant roots of the same developmental stage across different soils were highly similar, pinpointing to plant maturity as the primary driver of SM production. However, we found correlations and cooccurrence network connections between bacterial or fungal taxa with plant metabolites. This indicates the possibility of microbial induction of specific pathways leading to the synthesis of specific metabolites. Alternatively, these results could also pinpoint to the selection of specific microorganisms enriched by a specific metabolite. There are reports of distinct plant metabolic contents in different developmental stages. Chaparro et al. (11) reported plant developmental stage-dependent root exudation in *Arabidopsis thaliana*, which positively correlated with metabolism-specific rhizosphere microbial functions. Furthermore, soil and root microbial shifts were directly connected to metabolome-level changes in *Populus tremula* × *alba* (32). Here, we also found compounds in the roots known to be produced exclusively by microorganisms, like the mycotoxin citrinin (33).

In line with a recent publication characterizing bacteria isolated from *A. tinctoria* roots (27), we found some overlap between the culturable and unculturable *A. tinctoria* root microbiome. Rat et al. (27) isolated *Gammaproteobacteria* and *Alphaproteobacteria*, *Bacteroidetes*, and *Actinobacteria* as the most abundantly occurring root endophytes, and inoculation of a *Allorhizobium* strain led to significantly enhanced A/S concentrations in *A. tinctoria* hairy root cultures. This is in accordance with our results, as we identified bacterial rASVs (ASV 223 and ASV 194) of the *Allorhizobium-Neorhizobium-Pararhizobium-Rhizobium* genus group, which positively correlated with higher concentrations of acetylalkannin. The rASV 194 was also found in rhizosphere and bulk soil samples, suggesting that bacteria belonging to this ASV derive from the soil microbiome.

We found rASVs assigned to the genus group *Burkholderia-Caballeronia-Paraburkholderia* in positive correlation with alkannin (HPLC, Fig. 6) or in a negative correlation toward isovalerylalkannin (UHPLC-HRMS cooccurance network, Fig. S5 in the supplemental material) of *A. tinctoria*. The genus *Burkholderia* has been regularly described as a highly diverse and environmentally adaptable plant-associated group (34) prevalent in the rhizosphere or plant endosphere and was reported to cause metabolic changes in plants (35), for example, through their 1-aminocyclopropane-1-carboxylate (ACC) deaminase activity (36). Herpell et al. (37) isolated a *Paraburkholderia* sp. strain from the sweet potato phyllosphere that encoded several genes for plant hormone regulation, detoxification of xenobiotics, and aromatic compound degradation. Similarly, *Paraburkholderia phytofirmans* strain PsJN is a well-known plant colonizer that shows hormone and ACC deaminase production and is known to elicit plant responses (38, 39). Fungal elicitors, which enhance A/S production, have been previously described for the *Boraginaceae* species *Arnebia euchroma* (40, 41). In our study we found a moderate positive correlation between the root fungal rASV41 (*Penicillium jensenii*) and higher A/S levels (HPLC, Fig. 6).

We found one of the key metabolite nodes in the developmental stage network to likely be senecivernine N-oxide, a pyrrolizidine alkaloid that was recently reported in *Lithospermum erythrorhizon*, a *Boraginaceae* species (42). Pyrrolizidine alkaloids and their

N-oxides serve as plants' defense against herbivores and have been extensively studied (43, 44) due to their toxicity and potential for contamination of foods such as honey.

In the developmental stage subnetwork with the identified/annotated metabolites, the key metabolite of the network isovalerylalkannin had direct connections of positive correlations with four *Penicillium* species. Several species of this genus were already shown to affect plant growth and metabolism (45). In a study with citrus fruit, *Penicillium digitatum* infection induced the secondary and amino acid metabolism of the fruit (46). The strongest positive correlation ($R = 0.54$) to a direct neighbor of isovalerylalkannin in the network is a bacterial rASV from the genus *Labrys*. Plant rhizosphere isolates of this genus were previously described with indole acetic acid production and phosphate-solubilizing capabilities (47) and urea-decomposing and tryptophan-producing genes (48), which makes them potential plant growth-promoting strains. *Labrys* species were isolated several times from root nodules (49–51) but were not associated with nodulation and at present have unknown roles in the root nodule microbial community.

In accordance with previous publications (52), A/S derivatives in *A. tinctoria* at different developmental stages and grown in different soils consisted of >95% of the alkannin enantiomer.

Despite the increasing literature behind the metabolic content of *A. tinctoria* roots and the biological activities of A/S (7, 53, 54), not many interactions between specific chemical constituents of the plant have previously been discovered. In our study, we performed correlations between identified/annotated metabolites and found that two of the most abundant A/S in *A. tinctoria* roots, $\beta,\beta$-dimethylacrylalkannin (M1) and isovalerylalkannin (M2), exhibit moderate or weak correlations with abscisic acid ($R = 0.61$ and $R = 0.48$, respectively). Because the two mentioned A/S derivatives represented the majority of total A/S derivatives in *A. tinctoria* roots, their connection to abscisic acid is noteworthy. The potential role of abscisic acid in enhancing A/S yield in a production setting should be analyzed in more detail in a further study. Abscisic acid was shown to inhibit the growth of cultured cells of the *Boraginaceae* species *Onosma paniculatum* B. F., while it simultaneously reduced shikonin biosynthesis (55). However, our metabolomics data suggest this is not replicated in *A. tinctoria* roots; on the contrary, an increased abscisic acid concentration correlated with increased content of the most abundant A/S derivatives in the root system. With UHPLC-HRMS, we also annotated methyl jasmonate, which may also be produced by plant-associated microorganisms and which is known to induce metabolic changes in plants (56, 57).

Previous experiments have suggested the importance of developmental stages in identifying an ideal time point for harvesting plants for higher yield of the metabolites of interest (12–14). This knowledge was not available in the *in vivo* A/S production of *A. tinctoria*. We discovered that the fruiting developmental stage of *A. tinctoria* was the time point with the highest total A/S content under *in vivo* greenhouse conditions. Harvesting *A. tinctoria* roots at the fruiting stage in a production setting may represent an opportunity to make use of some of the other compounds with concentration peaks at the same stage, for example, abscisic acid, maltotetraose, 6-methoxyflavanone, ostruthol, and pyrogallol (Fig. 4A). Flavonoids, a compound class with anticancer, antiviral, and anti-inflammatory effects, among others (58), are represented in the analyzed roots by 6-methoxyflavanone, a metabolite reported as anxiolytic (59), a gamma-aminobutyric acid (GABA) allosteric modulator (60), and a bitter taste receptor inhibitor (61). Ostruthol, an anti-inflammatory coumarin described in *Peucedanum ostruthium* roots (62), is a potent acetylcholinesterase inhibitor with potential applications in treating Alzheimer's disease (63). Pyrogallol is a metabolite that was found to have antimicrobial properties (64). The biosynthesis of the above-mentioned metabolites would, due to their positive correlation with A/S, likely be positively affected by using certain microbes herein identified as beneficial toward A/S biosynthesis. The transition to the fruiting developmental stage also caused microbiome shifts, such as the abundance of rASVs in the root core microbiome, specifically higher abundances of *Allorhizobium-Neorhizobium-Pararhizobium-Rhizobium*, *Penicillium*, and *Trichoderma* associated with plants grown in the presence of

the Austrian soil microbiome, the appearance of high abundances of *Dyella*, *Myrothecium*, and *Chaetosphaeronema* in Greek A samples, and a sudden dominance of *Fusarium* species in the core of Greek B samples. The core microbiome is known as an important reservoir of microorganisms potentially influencing plant metabolism (21). Here, we analyzed core microbiome shifts through plant development together with metabolome shifts.

Our main hypothesis was that plant microbiota may influence plant metabolism and can enhance A/S production of *A. tinctoria* and/or that specific plant metabolites may specifically select or enrich certain microorganisms. Our findings support this hypothesis, as we successfully identified potential individual bacterial and fungal ASVs that correlated with higher A/S production. We observed core microbiome shifts, minor diversity changes in root, rhizosphere, and soil samples, and several significant correlations and network connections regarding metabolome-microbiome patterns in different stages of plant development. Such shifts have been identified to be connected to the peak of A/S biosynthesis, which is reached in the fruiting stage. This matches our hypothesis that different A/S concentration levels occur in different developmental stages, in line with a developmental stage-dependent colonization of plant microbiota.

Furthermore, we described previously unreported secondary metabolism constituents present in *A. tinctoria* roots and how their relative abundances change with plant development and also assessed how the identified and annotated compounds correlate to one another. We tested three different soils containing different microbiota for the cultivation of *A. tinctoria* resulting also in distinct root and rhizosphere microbiomes, however, not in significant changes in plant metabolite production. A comprehensive, untargeted metabolome analysis of *A. tinctoria* root extracts together with an untargeted microbiome analysis and Spearman correlations revealed microbial taxa correlating with the abundance of relevant A/S targeted.

Our findings can serve as a good basis for later multiomics studies and field experiments on *Boraginaceae* and other naphthoquinone-producing species. These taxa may be further explored for the involved mechanisms, both at the plant and microbial sides and for the development of microbe-based approaches to enhance A/S content in *A. tinctoria*. Such an enhancement would not only lead to increased production yields of A/S but also open the door toward using *A. tinctoria* roots as a starting material toward the isolation and production of metabolites positively correlated with A/S, calling for more studies on the feasibility of using the plant as a source of these compounds.

## MATERIALS AND METHODS

**Greenhouse experiment.** *Alkanna tinctoria* plants were provided as rooted acclimatized individuals that were originally collected and identified from natural populations and then produced by micropropagation from several mother plants by the Hellenic Agricultural Organization (HAO, Thessaloniki, Greece) in January 2018. The plants were transferred to 5-L pots containing 4.5 L of sterilized (121°C for 15 min) peat moss and perlite (volume ratio 2:1) mixed with 200 g of field soil collected either in Austria or in Greece. Thus, all plants were grown in a substrate with highly similar chemical and physical characteristics but that hosted those microbial communities prevailing in these three distinct soils. In nature, all three soils hosted various plant species related to *A. tinctoria* and belonging to *Boraginaceae*. The Austrian soil showed vegetation with *Echium vulgare*, whereas both Greek soils had *A. tinctoria* populations. Conductivity, pH, Ca, Fe, K, Mg, Mn, and P content of these soils were measured (Table S5 in the supplemental material) by extracting available soil nutrients in a chemical solution (0.5 M ammonium acetate/0.5 M acetic acid/0.02 M EDTA).

Plants were grown in the greenhouse with a 16-h light/8-h dark photoperiod at 25°C with 50% relative humidity (RH) and a photosynthetic photon flux density (PPFD) of 96 $\mu$molm$^{-2}$ s$^{-1}$. Plants were watered twice per week with deionized water and moved randomly once per week.

Plants were harvested at four different defined developmental stages; the first stage ("vegetative growth") was defined when more than 50% of the individuals started to produce new leaves, "blooming" was the stage when more than 50% of the individuals had flowers, and "fruiting" was the stage when more than 50% of the individual plants began to produce fruits. The final stage was "decay," when more than 50% of the plants had withered inflorescences. At each sampling stage of each soil, six individuals were harvested. Roots were cut at the root-stem transition and gently washed using tap water to eliminate the adhering debris. Rhizosphere samples were collected by washing roots in 45 mL of phosphate-buffered saline Tween 20 (PBST) solution and placed in a sonicator (Bransonic ultrasonic cleaner) for 10 min. The tubes were then centrifuged at 4°C at 3,500 rpm for 20 min after taking roots out, and pellets were collected by removing the supernatant (65). Corresponding bulk soil samples were also collected from each pot. The root system was divided into two equal parts; one part was freeze-dried and

shipped to Greece for SM analysis, while the other part together with the rhizosphere and bulk soil samples were kept at −80°C until performing DNA extraction. Fresh weights of roots and shoots were measured at each harvest point.

**Microbiome analysis. (i) DNA analysis and microbial community sequencing**. For extracting whole community DNA of roots, 100 mg of root material was freeze-ground by using 3.5-mm stainless steel UFO beads in a ball mill (Mixer Mill MM400, Retsch, Haan, Germany). The powdered material was then used for DNA extraction by using a modified cetyltrimethylammonium bromide (CTAB) protocol combined with a classical phenol-chloroform-isoamyl-alcohol extraction protocol (45). To get rid of additional PCR-inhibiting substances still inside the extracted solution, an additional DNA purification step with a DNeasy PowerClean cleanup kit (Qiagen) was applied on all samples. The rhizosphere and soil samples were processed with the FastDNA spin kit for soil and the FastPrep instrument (MP Biomedicals, Santa Ana, CA, USA) by following the instructions in the provided manual. For bacterial community analysis, we used the primer pair 799F-1175R (5′-AACMGGATTAGATACCCKG-3′ and 5′-ACGTCRTCCCCDCC TTCCTC-3′, respectively) (66) to target the V5-V7 region of the 16S rRNA gene. For fungal community analysis, an improved primer pair (67) amplifying the partial 5.8S ribosomal subunit gene 5.8S-Fun (5′-AACTTTYRRCAAYGGATCWCT-3′) and the whole internal transcribed spacer (ITS) 2 ITS4-Fun (5′-AGCCTCCGCTTATTGATATGCTTAART-3′) was used.

Paired-end sequencing (2 × 300 bp) was performed with a MiSeq reagent kit with v3 chemistry at a final loading concentration of 6 pM. The fungal ITS library was sequenced on a MiSeq system (Illumina, San Diego, CA, United States), whereas the bacterial 16S rRNA gene library was sequenced on the same Illumina platform by LGC Genomics (Berlin, Germany).

**(ii) Microbial data and cooccurrence network analysis**. Raw reads were filtered with Bowtie2 v.2.3.4.3 (68) to avoid the presence of Illumina's PhiX contamination, and quality was preliminarily checked with FastQC v.0.11.8. Primers were stripped using Cutadapt v.1.18 (69). Sequences were quality filtered, trimmed, and denoised, and amplicon sequence variants (ASVs) were generated with DADA2 v1.14 (70). Denoised forward and reverse ASV sequences were merged, and chimeras were removed. Filtered ASVs were checked using Metaxa2 v2.2.1 (71) and ITSx v1.1.2 (72) for targeting the presence of V5-V7 16S rRNA and ITS2 regions in archaeal and bacterial sequences and fungal sequences, respectively. Taxonomic assignment of the 16S rRNA gene ASVs and ITS-based ASVs was performed using the RDP classifier (73) against the SILVA v138 (74) database and UNITE 8.2 (75) database, respectively. BIOM objects with bacterial and fungal counts, respectively, were built and imported into the R-4.0.3 statistical environment for further analyses. Contaminating sequences were identified and eliminated by using the package decontam (76) by setting the threshold to 0.1 together with batch correlation to this particular experiment. Rare ASV filtering was applied with a 0.1% threshold in the RAM package filter.OTU function. The resulting tables were normalized and analyzed by using the vegan (77) rtk, emmeans, multcomp, dplyr, DeSeq2 (78), phyloseq (79), and mvabund packages. The 2,261 bacterial ASVs in root samples were assigned to 20 phyla, 43 classes, and 267 genera. In the rhizosphere, we identified 2,457 bacterial ASVs belonging to 19 phyla, 45 classes, and 298 genera. In bulk soil samples, we assigned 2,232 bacterial ASVs to 22 phyla, 46 classes, and 209 genera. Alpha-diversity values were calculated by adopting a multiple rarefaction method in which richness (observed ASVs) and diversity values (Simpson's diversity index [SDI]) were generated by averaging the results inferred after 999 rarefactions, starting from a minimum number of 1,422 and 2,632 for 16S rRNA gene and ITS data, respectively. ANOVA based on a linear model was calculated on the alpha-diversity values between groups, and significances were confirmed and investigated with a *post hoc* analysis (estimated marginal means). Beta-diversity Bray-Curtis distance matrices were created based on DeSeq2 count-normalized tables. Bray-Curtis values were analyzed in a pairwise PERMANOVA by using 999 permutations. A canonical analysis of principal coordinates (CAP) ordination plot based on the Bray-Curtis dissimilarity matrix was used to distinguish between the groups. Reproducibly occurring ASVs were calculated by first normalizing our data with DeSeq2's median of ratios method and creating a hybrid artificial factor by combining developmental stage and soil origin with the package dplyr and then using the core.OTU function from RAM with a 4/6 threshold, which calculated the ASVs present in 4 of 6 replicates in each artificial factor. Afterwards, to calculate the significantly differentially abundant rASVs, manyglm analysis (multivariate glm in mvabund package) was used considering both factors, followed by an ANOVA based on manyglm with 999 permutations.

We identified core microbiomes for each sample type (i.e., rhizosphere, root, and bulk soil), considering rASVs shared between soils or between developmental stage. In addition, we calculated the stable core microbiome according to the definition by Pfeiffer et al. (80) comprising rASVs across soils and developmental stages. For visualizing the core through Venn diagrams, venny 2.1.0 was used. The HPLC and liquid chromatography-mass spectrometry (LC-MS) metabolome data before correlation and network analysis were centered and scaled as a normalization method. Spearman correlations were done in the psych package, with false-discovery rate (FDR)-corrected $P$ values and correlation plots visualized with the ggcorrplot package. To create the cooccurrence networks, we used the DeSeq-normalized microbial and center-scaled metabolome data and inspected the data sets with t-distributed stochastic neighbor embedding (t-SNE) with the Rtsne package. R packages parallel, doParallel, caret, and rfPermute were used for performing random forest analysis on all the fungal, bacterial, and UHPLC-HRMS data. The significant features from the random forest analysis were submitted to Spearman correlation analysis in the psych package, and the correlation results were used to build undirected cooccurrence networks with the packages igraph, tidygraph, and ggraph. For creating box plots, bar plots, and ordinations, the ggplot2, ggvegan, reshape, and ggpubr packages were used.

**Metabolome analysis. (i) Chemicals and reagents**. Several standard compounds were used for metabolite identification and quantitation as previously described (26, 52). The following (purified by

column chromatography) standards were used: alkannin (Ikeda, Japan), shikonin (Ichimaru, Japan), acetylshikonin (ABCR GmbH, Germany), propionylshikonin (synthesized by Elias Kouladouros, Agricultural University of Athens, Greece), deoxyshikonin (TCI, Belgium), $\beta,\beta$-dimethylacrylshikonin (ABCR GmbH, Germany), and isovalerylshikonin (TCI, Belgium). For metabolite extraction and UHPLC-HRMS analysis, methanol (LC-MS-grade, Honeywell Riedel de Haën, USA) was used. To perform HPLC analysis, acetonitrile (HPLC-grade, Honeywell Riedel de Haën, USA), ultrapure water (Merck Millipore, Germany), and formic acid (HPLC-grade, Merck KGaA, Germany) were utilized. For UHPLC-HRMS analysis, formic acid (LC-MS reagent-grade, Honeywell Fluka, USA) was used. For hydrolyzing the alkannin/shikonin esters from root extracts to determine the alkannin:shikonin enantiomeric ratio, the following reagents were used: sodium hydroxide (Merck KGaA, Germany), hydrochloric acid (37%, Carlo Erba Reagents, France), and chloroform (ChemLab, Belgium). Hexane (HPLC-grade, Honeywell Riedel de Haën, USA) and 2-propanol (for liquid chromatography; Merck Millipore, USA) were used for chiral HPLC.

**(ii) Sample preparation for metabolite analysis**. Samples of *A. tinctoria* roots were stored at −80°C before being ground to a fine powder using a ball mill (Fritsch Pulverisette 0, Germany). Each powdered sample was weighed (70 mg) into microcentrifuge tubes, followed by extraction with 3 mL of methanol by ultrasound at 10% power for 3 h (Bandelin Sonorex Digital 10P, Berlin, Germany) and centrifugation for 10 min at 12,500 rpm (Hermle Z 216 MK, Wehingen, Germany). The supernatants were collected and subjected to UHPLC-HRMS and HPLC analyses after filtering with 0.22-$\mu$m syringe filters. To determine A/S enantiomeric ratio by chiral HPLC, the sample preparation procedure described by Tappeiner et al. (26) was followed.

**(iii) HPLC-UV/Vis method for the quantitation of A/S and determination of the alkannin: shikonin enantiomeric ratio**. To quantify the contents of alkannins/shikonins (each naphthoquinone derivative separately and total A/S content) in the methanolic root extracts prepared as described above, HPLC with UV-Vis detection set at 520 nm was used. All samples, regardless of growth stage or soil type, were analyzed. Analyses were performed on an Agilent 1200 HPLC instrument (Santa Clara, CA, USA) utilizing a VDSpher PUR C18-M-E 5-$\mu$m, 150 × 4.6 mm column (Berlin, Germany). The mobile phase was composed of ultrapure water (A) and acetonitrile (B). Each run lasted for 13 min with a flow rate of 1 mL/min, and samples were run in a randomized sequence to avoid bias. Elution was performed using the following solvent gradient: 0 min 30A/70B in 8 min to 100B and kept at that composition for 5 min. Before the next injection, the column was equilibrated for 5 min with the initial solvent composition. Column temperature was kept at 35°C. Quantitation of each A/S derivative using HPLC was performed by using calibration curves of A/S standards (shikonin, acetylshikonin, propionylshikonin, deoxyshikonin, $\beta,\beta$-dimethylacrylshikonin, and isovalerylshikonin). Data acquisition and processing of raw HPLC data were accomplished via the Agilent ChemStation software. Statistical analyses on the data set were performed using R v4.0.3 software.

The hydrolyzed root extract was analyzed by an adaptation of the normal phase HPLC method (26), with the detection wavelength set to 520 nm with the aid of a Chiralcel OD-H chiral column (Daicel, Osaka, Japan). The isocratic elution program consisted of *n*-hexane and 2-propanol in a 65:35 ratio, and elution time was 15 min with the mobile phase flow rate set to 0.8 mL/min. The retention time of each enantiomer was determined with A/S standards. The peak areas of alkannin and shikonin in hydrolyzed root extracts were estimated by integration through the HPLC software, and percent A:S enantiomeric ratio was estimated in each hydrolyzed extract. One sample of each soil type within each growth stage was randomly selected for hydrolysis and normal phase chiral HPLC analysis.

**(iv) UHPLC-HRMS method**. The UHPLC-HRMS data of the *A. tinctoria* root extracts were recorded on an LTQ Orbitrap Discovery (Thermo Scientific, Waltham, MA, USA) instrument. All samples from the vegetative growth, blooming, and fruiting stages were analyzed. The column used was an Acquity UPLC HSS C18 SB 1.8-$\mu$m, 2.1 × 100 mm column (Waters, Milford, MA, USA) kept at 50°C, with the mobile phase flow rate set to 0.3 mL/min. The solvents used were ultrapure water (A) and methanol (B), both with 0.1% formic acid added. The following gradient elution program was used: 0 min 95A/5B, 1 min 50A/50B, 8 min 0A/100B, 13 min 0A/100B, 13.01 min 95A/5B, and 16 min 95A/5B. Data were recorded in both positive and negative modes, with the capillary temperature set to 300°C using the MS/MS feature of the instrument. The MS/MS data were obtained for the six most intense *m/z* peaks in each full scan, with the normalized collision energy set to 35 eV. The acquisition and initial processing of the data were performed using Xcalibur (Thermo Scientific, USA) software. Data alignment and feature extraction were performed utilizing the XCMS online platform (The Scripps Research Institute, USA). Parameters used involved a maximum tolerated *m/z* deviation of 2.5 ppm in consecutive scans and a signal-to-noise threshold of 10. Retention time alignment was done with Obiwarp, while the maximum allowed shift in retention time was set to 5 s. The extracted data were passed through a filter, which removed features with a relative standard deviation in quality control (QC) samples of over 30% to avoid mistaking unreliable measurements with biological variation. The data were then normalized by median and underwent log transformation, resulting in normal distribution of the detected features. The identified/annotated features were subjected to correlation based on the Pearson coefficient. A hierarchical clustering heatmap of the normalized data set containing identified/annotated features was created using the Ward clustering method based on Euclidean distance. One-way ANOVA on the obtained data matrix containing all detected features was performed to estimate the number of statistically significant features present in the data set. The listed statistical analyses as well as the related data visualization were done with the help of MetaboAnalyst 5.0 (81) and R v4.0.3 software. Compound identification was performed by comparing *m/z* values, MS/MS fragmentation spectra, and retention times to those of commercially available A/S standards, while compound annotation was largely done by

matching *m/z* values and MS/MS fragmentation spectra to the mzCloud database by using Compound Discoverer 3.2 (Thermo Fisher Scientific, Waltham, MA, USA) while also taking into account retention times. For maximum hit confidence, matching was done by comparing our MS/MS spectra to database spectra obtained on reference compounds with Orbitrap systems by using the same collision-induced dissociation energy. Additional publicly available online databases, such as MassBank and PubChem, as well as available literature on studied compounds containing information on MS/MS fragmentation were used for metabolite annotation.

**Data availability.** The microbiome sequencing data were deposited in NCBI SRA and are available under the BioProject accession number PRJNA778946. The metabolome data obtained in this study are available at the NIH Common Fund's National Metabolomics Data Repository (NMDR) website the Metabolomics Workbench, where it has been assigned the project ID PR001283. The table of HPLC peak areas of A/S in all analyzed samples has been deposited to Zenodo and can be downloaded at https://doi.org/10.5281/zenodo.6349343.

## SUPPLEMENTAL MATERIAL

Supplemental material is available online only.
**FIG S1**, JPG file, 1.7 MB.
**FIG S2**, TIF file, 2.9 MB.
**FIG S3**, TIF file, 2.7 MB.
**FIG S4**, TIF file, 1.4 MB.
**FIG S5**, TIF file, 0.1 MB.
**TABLE S1**, DOCX file, 0.03 MB.
**TABLE S2**, DOCX file, 0.02 MB.
**TABLE S3**, DOCX file, 0.03 MB.
**TABLE S4**, DOCX file, 0.03 MB.
**TABLE S5**, DOCX file, 0.02 MB.

## ACKNOWLEDGMENTS

This project has received funding from the European Union's Horizon 2020 Research and Innovation Program under the Marie Skłodowska-Curie grant agreement number 721635. This work is also supported by NIH grant U2C-DK119886.

We thank the Hellenic Agricultural Organization (HAO) for providing initial plant material for the greenhouse experiment, Annalisa Cartabia and Alicia Varela Alonso for their help with the greenhouse experiment, the Large Laboratory Research Infrastructures and Instruments of the Department of Chemical Engineering, Aristotle University of Thessaloniki, and the Center of Interdisciplinary Research and Innovation of AUTh for access to UHPLC-HRMS and nuclear magnetic resonance (NMR) instruments, Evangelos Tzimpilis for technical support in operating the UHPLC-HRMS instrument, Helen Gika and George Theodoridis for advice and consultation related to UHPLC-HRMS methodology and giving access to a vibratory micromill for sample pulverization, and Christos Chatzidoukas for granting access to the HPLC-UV/Vis instrumentation. Furthermore, we thank Markus Gorfer for help and valuable consultation in designing the fungal PCR primers used for amplification of the fungal marker genes and Branislav Nikolić for technical help with the Illumina sequencing.

G.B., S.D., I.L., V.P.P., A.N.A., A.S., C.C., N.R., and Y.Z. designed the study. G.B., S.D., I.L., A.N.A., and A.S. supervised the experiments and writing and contributed to the discussion and interpretation of results. C.C., Y.Z., and I.L. helped to set up and manage the greenhouse trial and sample collection. C.C. and Y.Z. performed the microbiome-related laboratory works. N.R. and A.V. carried out the metabolomics-related experimental work and analysis on the metabolome data sets. L.A. performed the bioinformatic pipeline on the raw Illumina MiSeq data sets and provided valuable insights for the microbial ecology data analysis. C.C. performed the microbial ecology data analysis and microbiome-metabolome correlations. E.T. processed aboveground plant material for UHPLC-HRMS analysis, which, due to unfortunate sample loss, generated design imbalance and could not be displayed in the present study, but her tremendous work should be acknowledged in the form of authorship. C.C., N.R., Y.Z., and L.A. wrote the manuscript. All authors read and approved the final manuscript.

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
