## [Reviewer comments · mSystems]

Metabolite production in *Alkanna tinctoria* links plant development with the recruitment of individual members of microbiome thriving at the root-soil interface

Angela Sessitsch, Cintia Csorba, Nebojsa Rodic, Yanyan Zhao, Livio Antonielli, Günter Brader, Angeliki Vlachou, Evangelia Tsiokanos, Ismahen Lalaymia, Stéphane Declerck, Vassilios Papageorgiou, and Andreana Assimopoulou

Corresponding Author(s): Angela Sessitsch, AIT Austrian Institute of Technology GmbH

Review Timeline:

Submission Date:	May 16, 2022
Editorial Decision:	June 28, 2022
Revision Received:	July 8, 2022
Accepted:	July 17, 2022

Editor: Davide Bulgarelli

Reviewer(s): The reviewers have opted to remain anonymous.

Transaction Report:

DOI: <https://doi.org/10.1128/msystems.00451-22>

June 28, 2022

Dr. Angela Sessitsch
AIT Austrian Institute of Technology GmbH
Konrad-Lorenz-strasse 24
Tulln 3430
Austria

Re: mSystems00451-22 (Microbiome and metabolome development throughout the life-cycle of *Alkanna tinctoria* (L.) Tausch and specific associations between microbial taxa and secondary metabolites)

Dear Dr. Angela Sessitsch:

Thank you for submitting your manuscript to mSystems.

The processing of this revision has taken longer than anticipated as I was unable to recruit the three initial Reviewers who commented on your manuscript. Furthermore, the Reviewers provided me with contrasting opinions and I had to act as a tiebreaker.

First of all, I'd like to praise the effort the authors put in the revised version which I consider a significant improvement compared to the original submission. While the concern of Reviewer #1 is understandable, I am of the opinion that the experimental data presented and analysed in the revised version of the manuscript will intercept the interest of a broad scientific community and will likely fuel follow-up investigations.

I am therefore inclined to consider the revised version for publication. There are, however, a few points I'd like the authors to consider (besides the specific comments of the Reviewers below)

#1) I think you can be more explicit in the title and impact statement. For instance you could go with something along the lines 'Metabolite production in *Alkanna tinctoria* is driven by plant's development and is sufficiently resilient to microbiome variation across soils' or 'Metabolite production in *Alkanna tinctoria* links plant development with the recruitment of individual members of microbiome thriving at the root-soil interface'. Likewise, L. 61-62 how? I mean what kind of implications you have in mind? Target inoculation with rhizobia to trigger a differential metabolite production?

#2) L297-299. I believe is $R > 0.3$. Regardless, I'd justify the rationale for this choice as it is different from what reported in 284-286.

#3) L326-28. The soils used display very similar pH, it could be that in more acidic soil you may have different responses in the plants. I'd include a disclaimer here.

#4) L334-335: that's your closing for the importance statement!

#5) L386-388. 'Data available on request' is inadmissible. Either provide access to the data or remove the sentence altogether.

#6) I found Figure S5 more informative than Figure 7: have you considered swapping them?

#End from my side

Preparing Revision Guidelines

- Point-by-point responses to the issues raised by the reviewers in a file named "Response to Reviewers," NOT IN YOUR COVER LETTER.
- Upload a compare copy of the manuscript (without figures) as a "Marked-Up Manuscript" file.
- Each figure must be uploaded as a separate file, and any multipanel figures must be assembled into one file.
- Manuscript: A .DOC version of the revised manuscript

- Figures: Editable, high-resolution, individual figure files are required at revision, TIFF or EPS files are preferred

Sincerely,

Davide Bulgarelli

Editor, mSystems

Journals Department
Reviewer comments:

Reviewer #3 (Comments for the Author):

The manuscript has been extensively edited and improved with respect to the previous version. However, the main point arisen by this reviewer was not satisfied, since it is inherent to the experimental design. In particular the result description and comments are based on correlation analyses, well done, but which do not provide causation and/or mechanistic interpretation of the interplay between metabolites and microbiota.

Reviewer #5 (Comments for the Author):

The manuscript by Csorba et al. provide interesting evidence that growth-stage-dependent variation in specialized metabolite content co-occurs with changes in microbiome composition, suggesting a potential causal role of plant metabolites in modulating the microbiome (or vice versa). While the evidence for this is purely correlational, the study generates useful hypotheses for future study.

The previous comments of the reviewers have all been adequately addressed, and the manuscript has been significantly improved.

I just have a few small notes:

- There are some errors in the references in the submitted PDF: 'Error! Reference source not found.'
- The various figures seem to be using different fonts. I suggest making this consistent between the figures (all sans serif for example?). Italization of genera is also missing in figure 3, and (non-)capitalization of labels is not consistent across figures.

Response to reviewers

Editor

1. I think you can be more explicit in the title and impact statement. For instance you could go with something along the lines 'Metabolite production in *Alkanna tintoria* is driven by plant's development and is sufficiently resilient to microbiome variation across soils' or 'Metabolite production in *Alkanna tintoria* links plant development with the recruitment of individual members of microbiome thriving at the root-soil interface'. Likewise, L. 61-62 how? I mean what kind of implications you have in mind? Target inoculation with rhizobia to trigger a differential metabolite production?

Response: Thank you for taking the time to provide constructive feedback on the manuscript. The authors agree on changing the title of the study.

2. L297-299. I believe is $R > 0.3$. Regardless, I'd justify the rationale for this choice as it is different from what reported in 284-286.

Response: The mistake was corrected, and we clarified the reason behind the two different thresholds we used.

3. L326-28. The soils used display very similar pH, it could be that in more acidic soil you may have different responses in the plants. I'd include a disclaimer here.

Response: We added an additional sentence as a disclaimer in the manuscript.

4. L334-335: that's your closing for the importance statement!

Response: We thank you for your suggestion. We inserted this statement in our Importance section.

5. L386-388. 'Data available on request' is inadmissible. Either provide access to the data or remove the sentence altogether.

Response: We removed the sentence as requested. This small dataset of enantiomeric in the manuscript would not provide more clearance to the already displayed percentage, although it is part of the uploaded metabolome data in Zenodo.

6. I found Figure S5 more informative than Figure 7: have you considered swapping them?

Response: Indeed, swapping these figures provides a better view on the big networks in the main text.

Reviewer #3

The manuscript has been extensively edited and improved with respect to the previous version. However, the main point arisen by this reviewer was not satisfied, since it is inherent to the experimental design. In particular the result description and comments are based on correlation analyses, well done, but which do not provide causation and/or mechanistic interpretation of the interplay between metabolites and microbiota.

Response: We thank the reviewer for the comment on our work. Our experimental setup does not allow for a mechanistic explanation on the plant-microbe interface, and we are aware of the limitation of a correlation and network analysis based study. We still firmly believe such descriptive ecological studies involving both microbiome and metabolome are an important starting point for further and more detailed research on such an unusual plant.

Reviewer #5

The manuscript by Csorba et al. provide interesting evidence that growth-stage-dependent variation in specialized metabolite content co-occurs with changes in microbiome composition, suggesting a potential causal role of plant metabolites in modulating the microbiome (or vice versa). While the evidence for this is purely correlational, the study generates useful hypotheses for future study.

The previous comments of the reviewers have all been adequately addressed, and the manuscript has been significantly improved.

I just have a few small notes:

- There are some errors in the references in the submitted PDF: 'Error! Reference source not found.'

Response: The errors were due to the cross-referencing of figures and tables in the original document. We removed these in the updated version.

- The various figures seem to be using different fonts. I suggest making this consistent between the figures (all sans serif for example?). Italization of genera is also missing in figure 3, and (non-)capitalization of labels is not consistent across figures.

Response: We corrected the italization of the genera wherever we could and was needed on the figures, we also replotted everything so the fonts are uniform and have a sans-serif style.

July 17, 2022

Dr. Angela Sessitsch
AIT Austrian Institute of Technology GmbH
Konrad-Lorenz-strasse 24
Tulln 3430
Austria

Re: mSystems00451-22R1 (Metabolite production in *Alkanna tinctoria* links plant development with the recruitment of individual members of microbiome thriving at the root-soil interface)

Dear Dr. Angela Sessitsch:

Your manuscript has been accepted, and I am forwarding it to the ASM Journals Department for publication. For your reference, ASM Journals' address is given below. Before it can be scheduled for publication, your manuscript will be checked by the mSystems production staff to make sure that all elements meet the technical requirements for publication. They will contact you if anything needs to be revised before copyediting and production can begin. Otherwise, you will be notified when your proofs are ready to be viewed.

Please note that the last three lines of the Summary and Impact statement are identical. I believe this is a typo (as the authors agreed on a different conclusion for the impact statement): please rectify it at the proof stage.

Publication Fees:

If you would like to submit a potential Featured Image, please email a file and a short legend to msystems@asmusa.org. Please note that we can only consider images that (i) the authors created or own and (ii) have not been previously published. By submitting, you agree that the image can be used under the same terms as the published article. File requirements: square dimensions (4" x 4"), 300 dpi resolution, RGB colorspace, TIF file format.

We recognize that the video files can become quite large, and so to avoid quality loss ASM suggests sending the video file via <https://www.wetransfer.com/>. When you have a final version of the video and the still ready to share, please send it to mSystems staff at msystems@asmusa.org.

Sincerely,

Davide Bulgarelli
Editor, mSystems

Journals Department
Table S3: Accept
Fig. S4: Accept
Fig. S3: Accept
Fig. S1: Accept
Table S2: Accept
Table S1: Accept
Fig. S5: Accept
Table S4: Accept
Fig. S2: Accept
Table S5: Accept